# The Biological Context of C-Reactive Protein as a Prognostic Marker in Renal Cell Carcinoma: Studies on the Acute Phase Cytokine Profile

**DOI:** 10.3390/cancers12071961

**Published:** 2020-07-19

**Authors:** Helene Hersvik Aarstad, Gigja Guðbrandsdottir, Karin M. Hjelle, Leif Bostad, Øystein Bruserud, Tor Henrik Anderson Tvedt, Christian Beisland

**Affiliations:** 1Department of Clinical Science, Faculty of Medicine, University of Bergen, N-5020 Bergen, Norway; Helene.Aarstad@uib.no; 2Department of Urology, Haukeland University Hospital, N-5021 Bergen, Norway; gigja.gudbrandsdottir@helse-bergen.no (G.G.); karin.margrethe.hjelle@helse-bergen.no (K.M.H.); christian.beisland@helse-bergen.no (C.B.); 3Department of Clinical Medicine, Faculty of Medicine, University of Bergen, N-5020 Bergen, Norway; 4Department of Pathology, Haukeland University Hospital, N-5021 Bergen, Norway; bostadleif@gmail.com; 5Department of Medicine, Section for Hematology, Haukeland University Hospital, N-5021 Bergen, Norway; tor.henrik.anderson.tvedt@helse-bergen.no

**Keywords:** renal cell carcinoma, acute phase reaction, C-reactive protein, IL33Rα, IL1 family, IL6 family, tumor necrosis factor α

## Abstract

High serum levels of the acute phase protein C-reactive protein (CRP) are associated with an adverse prognosis in renal cancer. The acute phase reaction is cytokine-driven and includes a wide range of inflammatory mediators. This overall profile of the response depends on the inducing event and can also differ between patients. We investigated an extended acute phase cytokine profile for 97 renal cancer patients. Initial studies showed that the serum CRP levels had an expected prognostic association together with tumor size, stage, nuclear grading, and Leibovich score. Interleukin (IL)6 family cytokines, IL1 subfamily mediators, and tumor necrosis factor (TNF)α can all be drivers of the acute phase response. Initial studies suggested that serum IL33Rα (the soluble IL33 receptor α chain) levels were also associated with prognosis, although the impact of IL33Rα is dependent on the overall cytokine profile, including seven IL6 family members (IL6, IL6Rα, gp130, IL27, IL31, CNTF, and OSM), two IL1 subfamily members (IL1RA and IL33Rα), and TNFα. We identified a patient subset characterized by particularly high levels of IL6, IL33Rα, and TNFα alongside an adverse prognosis. Thus, the acute phase cytokine reaction differs between renal cancer patients, and differences in the acute phase cytokine profile are associated with prognosis.

## 1. Introduction

Renal cell carcinoma is a common malignancy and among the most lethal genitourinary cancers [1]. Standard treatment involves partial or radical nephrectomy for local tumors, whereas targeted therapies can be considered for metastatic disease [2,3]. The systemic serum levels of several cytokines, especially Interleukin (IL)6 are associated with prognosis in several urogenital cancers [4,5,6], including renal cell carcinoma [7,8]. IL6 belongs to the IL6 cytokine family. These cytokines utilize glycoprotein 130 (gp130) for intracellular signal transduction [9]. IL6 along with other family members are regulators of the acute phase reaction and initiate intracellular signaling either through the cytokine binding to the complete membrane receptor (classical signaling) or through the binding of the soluble cytokine-receptor complex to membrane-expressed gp130 (trans signaling) [9]. Thus, IL6 family cytokines form an interacting network of soluble mediators, including the cytokines themselves together with their membrane-bound and biologically active soluble receptor chains.

The acute phase reaction is a systemic response that accompanies acute and chronic inflammation. It is triggered by tissue damage and characterized by the altered serum levels of several inflammation-regulatory proteins, including C-reactive protein (CRP), and can be induced by IL6 family cytokines, as well as IL1β and tumor necrosis factor (TNF)α [10,11]. CRP binds a wide range of exogenous and endogenous ligands; these complexes bind to Fc or CD14/Toll like receptors (TLRs), thereby leading to a systemic plasma protein response involving several cytokines that reinforce the initial CRP-inducing cytokine response [11]. IL6 is important for the release of several acute phase proteins. The overall acute phase protein profile differs between various inducers, and other IL6 family cytokines have effects similar to IL6 [10,12].

IL1 α/β are members of the IL1 subfamily of the IL1 cytokine family and are important in the acute phase reaction together with the soluble IL1 receptor antagonist (RA) [13,14,15,16]. The release of IL1RA by hepatocytes as part of the acute phase reaction is at least partly regulated by IL6 [17].

IL33 is another member of the IL1 subfamily of cytokines, and the soluble IL33 receptor α chain (IL33Rα) should also be regarded as an acute phase protein [18,19,20,21]. IL33 binds to IL33Rα, which forms a dimer with the signal-initiating IL1RAcP co-receptor [22]. The same co-receptor is utilized by the IL1 receptor chain [22]. The soluble IL33 receptor IL33Rα (sIL33Rα) is a decoy receptor that shows altered systemic levels in several diseases [23,24] and is identical to the extracellular region of the membrane-bound (referred to as IL33Rα long or IL33RαL) chain, except for five additional amino acids [25,26,27,28]. A third IL33RαV variant, with another hydrophobic tail and lacking one extracellular domain, also exists [29]. IL33RαL is expressed by various cells, including epithelial, endothelial, and immunocompetent cells [25,30]; IL33RαV is expressed by certain epithelial and immune cells [25]; sIL33Rα is released by several cells, including kidney and immunocompetent cells [25]; and IL33 is expressed mainly by non-hematopoietic cells [30]. Downstream receptor signaling involves MyD88 and several of its downstream pathways that ultimately target NF-κβ and AP-1 [30], but IL33 can also bind to chromatin or directly inactivate NF-κβ [30,31,32,33]. Its final effect seems to be the stimulation of renal carcinogenesis [27].

Tumor diameter [34] and preoperative serum CRP levels are independent prognostic parameters in non-metastatic renal cell carcinoma; CRP thus serves as a marker of the acute phase reaction [35]. The aim of the present study was to characterize the heterogeneity of the cytokine-driven acute phase reaction (i.e., the biological context of CRP) in patients with renal cell carcinoma by investigating an extended pre-therapy acute phase cytokine profile that includes seven IL6 family members, IL1 subfamily members (IL33Rα and IL1β/IL1RA), and TNFα.

## 2. Results

### 2.1. Clinical, Biological, and Prognostic Characteristics of the Renal Cancer Patients

During a defined time period, 154 patients were surgically treated for renal cancer. They all gave their written informed consent but due to practical or technical reasons, a preoperative serum sample could be collected only for 118 patients. These 118 patients included 9 with metastatic and 109 with local disease. Our hospital is responsible for the treatment of all renal cancer patients for a defined geographical area, and our patient cohort represents all diagnosed patients from a defined time period. The characteristics of the whole patient cohort and for the patients only with non-metastatic disease are presented in Table 1, whereas the characteristics of the patients who could not be sampled preoperatively due to practical or technical reasons are given in Appendix A.

The whole patient cohort included nine patients with detectable metastatic disease at the time of diagnosis. The 109 patients with non-metastatic disease included 80 surviving patients, 11 patients who died from relapsed cancer, and 18 patients who died from other causes. The IL33Rα levels were determined for 96 patients and the other nine cytokines for 97 patients (one additional patient); six patients with metastatic disease were included for all the mediators. These 97 patients comprise of all patients who were sampled during the study period without additional selection. Our cohort included 70 survivors, six patients who died from their metastatic cancer disease detected at the time of diagnosis, 7 additional patients who also died from their renal cell carcinoma, and 14 patients who died from other causes.

We compared the clinical and biological parameters listed in Table 1 for potentially cured patients (i.e., no detectable metastases at the time of diagnosis) with those of cancer-free survivors and patients who later died from relapse/metastases. These last two groups differed significantly from the cancer-free survivors with regard to their serum CRP levels (*p* = 0.003), frequency of large tumors at the time of diagnosis (*p* < 0.001), and frequency of Fuhrman G3-G4 nuclear grading (*p* = 0.001). All these parameters are regarded as prognostic factors for renal cancer patients, and these differences are, therefore, expected [38,39,40,41,42]. Thus, these patient characteristics show that our cohort of renal cancer patients can be regarded as representative. The patients included in our cytokine studies were randomly selected from the 118 patients in the cohort.

### 2.2. The CRP Levels in Renal Cancer Patients; Strongest Associations with Tumor Characteristics, Weak Associations with Comorbidity, and Only Associated with IL6 among the Ten Cytokine Mediators

The acute phase reaction can be initiated by inflammation and tissue damage, but epidemiological studies have also demonstrated that the CRP levels in elderly individuals can be associated with frailty or comorbidity, i.e., they can be a part of the aging process [43,44,45,46]. We thus investigated whether the CRP level at the time of diagnosis was significantly associated with clinical characteristics, tumor characteristics, comorbidity scores, or cytokine serum levels (Appendix A). The preoperative CRP levels showed the strongest associations with tumor stage (i.e., diameter; Kendall’s τ 0.315) and the presence of necrosis in the tumor (Kendall’s τ 0.332). The Eastern Cooperative Oncology Group (ECOG) performance status showed an association of borderline significance, whereas the Charlson comorbidity index and the American Society of Anesthesiologists (ASA) physical classification score showed no associations. Thus, the CRP level mainly reflected the characteristics of the malignant disease among the patients.

Preoperative serum CRP levels showed a correlation with IL6 levels (Kendall’s τ 0.301, *p* < 0.001), whereas no significant correlation with CRP was seen for the IL1 subfamily mediators IL33Rα (Kendall’s τ 0.173) or IL1RA (Kendall’s τ 0.246). For the other IL6 family members and TNFα, the Kendall’s τ value was generally lower (usually < 0.10) and/or associated with *p*-values > 0.10. IL6 is regarded as a major driver of the acute phase reaction [10], and an association between the CRP and IL6 levels is, therefore, not unexpected. The systemic IL1β levels were generally low with minor variations and undetectable levels in several patients; the detection of low IL1β levels is consistent with previous studies of cancer patients [47]. Thus, high IL6 levels are an additional phenotypic characteristic of the acute phase reaction for renal cancer patients with high CRP levels, whereas variation in the other acute phase cytokine mediators is not reflected by CRP in renal cancer patients.

### 2.3. Serum Levels of the IL1 Subfamily Mediators IL33Rα and IL1RA Show No Significant Correlation; Only IL33Rα Is Increased in Metastatic Disease, and Only IL33Rα is Associated with Survival

The preoperative serum levels of IL33Rα and IL1RA did not show any significant correlation. We also classified the patient subset with non-metastatic disease into three groups based on their IL33Rα/IL1RA levels: (i) both with levels above the corresponding median; (ii) only one of the mediators having a level above the median; and (iii) both levels below the corresponding median. These three patient subsets did not differ in their overall survival. Finally, neither the IL33Rα nor IL1RA serum levels showed any significant associations with ECOG performance status, ASA score, Charlson comorbidity index, tumor size, or Fuhrman nuclear grading.

The IL33Rα levels were significantly higher for patients with metastases (*n* = 6, median level 29,130 pg/mL, range 23,520–162,569 pg/mL) compared to the patients with non-metastatic disease (*n* = 90, median 22,656 pg/mL, range 7053–75,572 pg/mL, Wilcoxon’s rank sum test, *p* = 0.017). We also classified our patients with non-metastatic disease based on their tumor stage. The IL33Rα levels for patients with large tumors (i.e., diameters > 7 cm) differed significantly from patients with metastatic disease (*p* = 0.038) but not from the patients with non-metastatic disease and small tumors (Figure 1). In contrast, the IL1RA levels for patients with metastatic disease (*n* = 6, median 802 pg/mL, range 335–1607 pg/mL) did not differ significantly from those of patients without metastases (*n* = 91, median 684 pg/mL, range 281–2711 pg/mL), and IL1RA also did not differ between patients with non-metastatic disease and those with metastatic disease or between those with small versus large tumors without metastases. Thus, IL33Rα and IL1RA belong to the same IL1 cytokine subfamily and should both be regarded as acute phase mediators. Nevertheless, these two mediators differ in metastatic versus non-metastatic disease and thereby contribute to the heterogeneity of the acute phase cytokine reaction in renal cancer patients.

We thus investigated the association between survival and the IL33Rα level, CRP level, Leibovich score, tumor size, Fuhrman’s nuclear grading, ASA score, and age via univariate Cox prediction analyses. We then examined the death from renal cancer and overall survival for the patients who were classified as radically treated after surgery (Table 2). IL33Rα showed an association of borderline significance with cancer-related death, whereas highly significant associations were observed for the tumor characteristics and CRP levels. For overall survival, significant associations were seen for the tumor characteristics, serum CRP, and patient age, whereas IL33Rα did not reach significance. Finally, IL1RA showed no significant associations with cancer-related death or overall survival in the Kaplan–Meier or Cox analyses.

The Leibovich score is used for the prognostic evaluation of patients with renal cancer [38,40,42,48]. We, therefore, investigated the IL33Rα levels and Leibovich score using a multivariate analysis for patients with a clear-cell subtype of kidney malignancy. We had relatively few cancer-related deaths in our cohort, and for this reason, we included only these two parameters. Moreover, the Leibovich score was chosen because it includes several prognostic parameters, and the IL33Rα level remained significant when corrected for the Leibovich score (Table 3).

### 2.4. The IL6 Cytokine Family Profile Identifies Patient Subsets That Differ in the Prognostic Impact of IL33Rα, Whereas the Impact of IL1RA/TNFα Does Not Differ

We investigated the serum levels of the IL6 family cytokines IL6, IL27, IL31, CNTF, and OSM, together with the soluble receptor components gp130 and IL6Rα. These IL6 family mediators form an interaction network through their overlapping receptor binding (with gp130 as a common signal-initiating receptor chain), common downstream intracellular signaling, and the potential for both classical and trans signaling (i.e., binding of the soluble receptor/ligand complex to membrane-expressed gp130) for several of these cytokines [9]. The overall results were investigated by hierarchical clustering analysis (Figure 2). CNTF and IL6 had the widest variation ranges among the included mediators. This analysis identified two main patient subsets that did not differ with regard to the serum levels of the IL6 family mediators, IL1 subfamily mediators, TNFα, or CRP. Finally, the number of patients dying from renal cancer (i.e., patients with metastases at diagnosis or later relapse) or dying from other causes did not differ between the two main patient clusters.

Each of these two main clusters was further divided into two subclusters characterized mainly by differences in their IL6 and CNTF levels, as indicated to the right in Figure 2. Patients included in the two sub-clusters are characterized by low or relatively low levels of IL6 and/or CNTF (Figure 2 right part, indicated by the blue color in the figure and referred to as IL6^low^CNTF^low^ patients). We first compared the soluble mediator levels for the IL6^high^CNTF^high^ and IL6^low^CNTF^low^ patients (Appendix A). The systemic IL1RA levels were significantly higher for the IL6^low^CNTF^low^ patients (median 736 pg/mL, range 371–2710, Wilcoxon’s test, *p* = 0.027) compared to the IL6^high^CNTF^high^ patients (656 pg/mL, range 280–1493). The systemic levels of CRP, TNFα, and other IL1 subfamily or IL6 family mediators did not differ significantly between these two patient subsets. Lastly, the number of patients dying from renal cancer (i.e., patients with metastases at diagnosis or later relapses) or dying from other causes also did not differ between the IL6^high^CNTF^high^ and IL6^low^CNTF^low^ patients.

We used Kaplan–Meier analyses to compare the associations between IL33Rα levels and cancer-related death (metastases or relapse) for the IL6^low^CNTF^low^ and IL6^high^CNTF^high^ patients (see Figure 2). Patients were classified into quartiles based on the IL33Rα variation range. Patients in the three lower quartiles showed a similarly low mortality for both the IL6^low^CNTF^low^ and IL6^high^CNTF^high^ subsets and were, therefore, classified together and compared with the patients in the highest quartile. The results are presented in Figure 3. A significant association between prognosis and IL33Rα levels was only observed for the IL6^low^CNTF^low^ patients, whereas such an association was not detected for the IL6^high^CNTF^high^ patient subset. Thus, the prognostic impact of a single acute phase mediator (i.e., IL33Rα) may differ between patient subsets identified by the acute phase cytokine profile (i.e., the IL6 family profile). This prognostic impact only for certain patients may also explain why IL33Rα levels did not differ when comparing the survivors and non-survivors in our whole study population (Appendix A).

### 2.5. The Prognostic Impact of an Extended Acute Phase Cytokine Profile for Renal Cancer Patients

Our IL6 family cytokine profiling (Figure 2) clearly illustrates that the acute phase reaction in patients with renal cell carcinoma possessed heterogeneity that was only partly reflected in the CRP level. To further investigate the prognostic impacts of these differences on the acute phase profile, we performed a hierarchical clustering analysis based on TNFα, two IL1 subfamily mediators (IL1β was not used due to undetectable levels in many patients and only minor variations between patients), and the seven IL6 family members. Our present and previous studies suggest that IL33Rα is associated with the acute phase reaction. Moreover, previous studies have shown that the nine other mediators are involved in the regulation of the acute phase response (see Section 1). The results of this clustering analysis are shown in Figure 4. After this analysis, two main patient subsets were identifiable.

We performed a Kaplan–Meier survival analysis comparing the two main subsets identified in Figure 4. This analysis is presented in Figure 5. As shown, the two patient clusters differed in their disease-specific survival. As expected from the results presented in Figure 4, the two main patient clusters did not differ in their overall survival, indicating that most of the patients (14 patients) died from other cases, and only 13 patients (six with metastases at the time of diagnosis) died from their malignant disease.

We ultimately compared the serum levels of all acute phase proteins, including the CRP levels, between the two main subsets identified (Table 4). The two subsets showed highly significant differences in their IL6, IL33Rα, and TNFα levels, whereas their IL1RA and CRP levels showed differences of only borderline significance. The IL6 and IL33Rα differences remained significant even after Bonferroni corrections. Thus, the two main patient clusters were mainly determined by the levels of the three mediators, and this sub-classification, therefore, was determined by acute phase characteristics that are only partly reflected in the serum CRP levels.

## 3. Discussion

The serum CRP level is a generally accepted prognostic factor for patients with renal cell carcinoma [41,49]. However, CRP is only one of several acute phase proteins, and the systemic serum profiles of acute phase cytokines (i.e., potential drivers of the acute phase reaction) seem to differ between patients and may also depend on the cause of the acute phase reaction [10]. In this context we investigated the acute phase cytokine profiles among a large group of patients with renal cancer admitted for surgical treatment.

As described above, our original cohort of 118 patients (109 without metastases) represents an unselected group of patients, i.e., the patients were derived from a defined geographical area during a defined time period and included all patients that could be sampled before surgery. The 97 patients included in our present study were randomly selected from this cohort. For this reason, we regarded our patients to be representative. Only a minority of the patients had advanced disease, while a majority of the patients had stage T1 tumors and tumor-node-metastasis (TNM) stage I (see Appendix A). As expected, the cancer-free survival was high, but due to the high median age, several patients died from causes other than their cancer.

We cannot exclude the possibility that inflammaging (i.e., inflammation associated with aging, see [50]) or other chronic inflammatory diseases contributed to the observed acute phase reaction. However, we did not perform any additional selection of patients included in our present cytokine studies. Our present results should, therefore, be regarded as real-word data from a representative group of patients with renal cancer. We cannot exclude the possibility that the acute phase reaction in some of our patients may have been, at least partly, caused by inflammaging or nonmalignant chronic inflammatory diseases, but, despite this, we still detected a prognostic impact of the acute phase cytokine response when investigating our unselected patients. None of the survival analyses demonstrated different results with the inclusion of age, Charlson comorbidity index, ASA score, and ECOG performance status as co-variates.

Recent studies show that CRP is an important regulator in inflammation, but in clinical practice, it is used as a marker of both inflammation and the complex acute phase reaction [11]. The aim of the present study was to investigate the biological context of CRP (i.e., the acute cytokine network response and the acute phase reaction) in a representative cohort of patients with renal cancer. Our selection of mediators was based on previous studies showing that IL1, TNFα, and IL6 are important in the development of the acute phase reaction [10]. First, the IL1 cytokine family includes the IL1 subfamily [22] with the members IL1α/β and IL33, their receptors, and the antagonistic IL1RA. The IL1 and IL33 binding receptor chains co-localize with the same signal-initiating IL1RAcP co-receptor [22]. We, therefore, included IL1β together with its antagonist IL1RA in our present study [10,11,22]. In addition, we included the soluble IL33 decoy receptor IL33Rα because this biomarker should also be regarded as an acute phase protein (i.e., a systemic marker of inflammation) [18,19,20,21], but we did not include IL33 itself because it is produced by renal cancer cells and its local release is likely more important [51]. Second, we included TNFα, which is an acute phase cytokine and also important for the development of the acute phase reaction [10]. Finally, we investigated the levels of IL6 family members and soluble IL6 receptor components because IL6 is an important regulator of the acute phase reaction [10], and the systemic IL6 level also seems to have a prognostic impact on renal cancer [52,53,54]. We focused on the IL6 cytokine family profile because several such family members contribute to the regulation of the acute phase response. Their receptor binding partly overlaps, their intracellular signaling is similar, and several of them show both classical and trans signaling [9,10]. We then included IL6 family members that show systemic levels in a majority of immunocompetent and immunocompromised individuals [55,56].

We investigated a relatively large group of renal cancer patients that were randomly selected from a consecutive group of patients. Our patients should be regarded as representative in their clinical (e.g., age, performance status, and survival), biological (e.g., tumor and cancer cell characteristics), and prognostic parameters (e.g., Leibovich score, tumor characteristics, and CRP level). However, our patients had a long follow-up time. For this reason, Fuhrman nuclear grading was used at the time of inclusion instead of the newly recommended system [41,57].

The prognostic impact of CRP shows that the acute phase reaction is important in renal cell carcinoma [39,58,59,60]. CRP is not only a marker but also a mediator with distinct biological functions [61,62,63]. However, the systemic acute phase reaction is a very complex response, and the aim of our present study was, therefore, to investigate the systemic levels of acute phase cytokines in renal cancer patients with a focus on the acute phase cytokine profiles, rather than those of single cytokines. IL6 family cytokines were included because they are important regulators of the acute phase reaction [10], but we investigated only IL6 family cytokines that usually show detectable serum levels [55,56]. IL1β/IL1RA and TNFα are important in the regulation of the acute phase response [10]. The inclusion of IL33Rα in our acute phase cytokine profile is justified by our present results, describing an association between IL33Rα levels and prognosis, and by those of previous studies showing that IL33Rα is an acute phase protein [18,19,20,21].

Studies on several malignancies (including renal cancer) suggest that the IL33/IL33Rα axis could be important in tumorigenesis through exerting direct effects on malignant cells [64] or indirectly through effects on stromal cells [65], including altering the regulation of tumor angiogenesis [66]. An association between serum IL33Rα levels and prognosis has been described for patients with hepatocellular carcinoma [67] and breast cancer [68]. A recent study also investigated serum IL33 levels and tumor IL33 expression via immunocytochemistry for renal cancer patients [27]. A high tumor expression of IL33 was then associated with advanced disease and an adverse prognosis; additional experimental studies showed that IL33 enhanced cancer cell growth and induced chemoresistance. A similar prognostic impact was described in another retrospective study that also assessed IL33 tumor expression via immunohistochemical staining [51]. However, yet another study described an adverse prognostic impact from the low renal cancer expression of IL33 at the mRNA level [26]. The use of different methodological approaches may explain this discrepancy. We also observed a possible prognostic impact of sIL33Rα independent of the Leibovich score, which is mainly based on tumor characteristics [38], but the low number of cancer-related deaths represents a limitation for the statistical analysis of patient survival in our present study. Finally, the immunoregulatory functions of IL33/IL33Rα may also be important for the effect of this axis on human malignancies, e.g., through induction of Treg cells or the inhibition of antigen presentation [26,33,69,70]. For these reasons and because of the similarities in downstream receptor signaling between IL1 and IL33, we included sIL33Rα in our acute phase cytokine profile together with IL1β and IL1RA. This was further supported by previous studies showing that IL33Rα is associated with prognosis in renal cancer and represents a systemic marker of inflammation [71].

We investigated IL6 family members that have detectable serum levels in most healthy individuals [55,56]. IL6 family cytokines have similarities in their receptor structures, with gp130 being the common signaling structure for all the receptors; in addition, some of the receptors can bind different IL6 family cytokines, and several family members are capable of both classical and trans signaling. For this reason, one should regard this family as the IL6 family network. We, therefore, focused on the IL6 family profile rather than on single family members. Even though several of these members seem to be involved in regulating the acute phase response, differences in the IL6 family profile could be used to identify patient subsets by hierarchical clustering analyses. However, the main subsets identified by hierarchical clustering based on IL6 family cytokines showed no association with CRP levels or patient survival (Figure 2). Finally, even though IL6 and CRP levels showed a significant correlation, the levels of these two cytokines did not differ when comparing the two main patient subsets (i.e., IL6^high^CNTF^high^ versus IL6^low^CNTF^low^ patients, see Figure 2). This is likely due to the impact of other IL6 family members (especially CNTF) in this IL6 family-based cluster analysis. This is also consistent with our observation that IL6 is the only cytokine biomarker presenting a significant correlation with CRP levels.

We ultimately performed a hierarchical clustering analysis based on TNFα, two IL1 subfamily members, and seven IL6 family members. Based on this overall acute phase cytokine profile, we identified two main subsets. These two subsets were not independent of the CRP level but differed significantly with regard to patient survival. The majority of patients dying from their malignant disease (metastatic disease at the time of diagnosis, later death from relapse) were included in a cluster characterized by especially higher levels of IL6, IL33Rα, and TNFα compared to the other main cluster, whereas IL1RA and CRP only showed differences with borderline significance. The IL6 and IL33Rα differences remained significant even after Bonferroni corrections. Thus, the overall clustering analysis based on an acute phase profile identified two main subsets. The patient survival differed between these two subsets, and this prognostic impact mainly reflected differences in IL6/IL33Rα/TNFα.

## 4. Materials and Methods 

### 4.1. Patients

This retrospective biobank study was approved by the regional ethics committee (REK VEST 78/05) and the Norwegian Social Science Data Services; the study was conducted in accordance with the Declaration of Helsinki. Blood samples were collected after written informed consent from 118 consecutive patients with newly diagnosed renal cell carcinoma during the time period of 2007–2010 (median observation time 100 months, range 4–120 months). All patients were followed according to our risk-stratified follow-up program for surgically treated renal cell carcinoma [72]. The present study included 97 randomly selected patients from this cohort. The patient and tumor characteristics are presented in Table 1 and Appendix A.

### 4.2. Analyses of CRP Levels

CRP levels were analyzed using the immunoturbidimetric method provided by Roche (Basel, Switzerland). During the entire period, the lower limit of detection for the serum CRP was 1 mg/L.

### 4.3. Blood Sampling and Cytokine Analyses

Peripheral venous blood samples were collected on the morning of the day of the planned renal cancer surgery. Samples were stored at room temperature for less than two hours before they were centrifuged. The serum was collected, aliquoted, and later stored frozen at −80 °C until being analyzed.

Samples derived from the 97 unselected patients were available for analyses. The samples were then thawed and centrifuged at 16,000× *g* for 4 min immediately before analysis. The IL6 levels were analyzed by a high-sensitivity ELISA kit (R&D Systems Europe Ltd., Abingdon, UK). Gp130, IL6Rα, IL27, IL31, OSM, IL1RA, and TNFα were determined using a Human Premixed Multi-Analyte Kit for Luminex technology (R&D Systems). IL33Rα was also determined using Luminex analyses (R&D Systems). A Human Pituitary Magnetic Bead Panel 1 was used to measure CNTF (EMD Millipore Corporation, Billerica, MA, USA). All analyses were performed strictly according to the manufacturer’s instructions and the levels estimated by using a Luminex^®^ 100^TM^ (Luminex Corporation, Austin, TX, USA). All results are presented as the mean level of duplicate determinations.

One patient sample was included in all assays to evaluate the inter-platelet variation, but we did not detect any substantial differences between assays. The variation between duplicates was generally less than 10% of the mean concentration. Neither IL33Rα nor IL1RA levels showed any correlations with the sample storage time.

### 4.4. Statistical and Bioinformatical Analyses

The IBM^®^ SPSS^®^ Statistics software, version 25.0 (IBM Corp., Armonk, NY, USA), was utilized. A comparison of descriptive data was performed using cross-tables and an exact Chi-square test. A Mann–Whitney U test was used for a comparison between different groups, and Kendall’s tau (τ) was used for correlation analyses. Kaplan–Meier analyses were used for the percentage estimation of outcome prediction, including a Log-Rank test between groups. Cox proportional hazard models were also used for survival analyses. A *p*-value < 0.05 was regarded as statistically significant. Correction for multiple comparisons was done by Bonferroni. Bioinformatical analyses were performed using J-Express (MolMine AS, Bergen, Norway) [73]. All cytokine and receptor levels were normalized by their median values, naturally log-transformed, entered into a complete linkage, and used to generate hierarchical clustering. The distance measures were Euclidean.

## 5. Conclusions

The systemic levels of the acute phase protein CRP are a generally accepted prognostic factor for renal cell carcinoma. Our present study shows that the acute phase cytokine profile differs between renal cancer patients, and most cytokine serum markers included in our present study showed no association with serum CRP levels. Based on differences in the overall acute phase cytokine profile, we classified renal cancer patients into two main subsets that differed significantly with regard to prognosis. Our results suggest that the possible prognostic impact of an extended acute phase cytokine profile or acute phase proteins other than CRP depends on biological context and differs between patient subsets. The possible prognostic impact of the acute phase cytokine profiles should be further investigated for patients with renal cell carcinoma. However, the cancer-related patient death was relatively low in our patient cohort, and the possible prognostic impact of these phenotypic differences in the acute phase reaction has to be further investigated in larger patient cohorts.

## Figures and Tables

**Figure 1 cancers-12-01961-f001:**
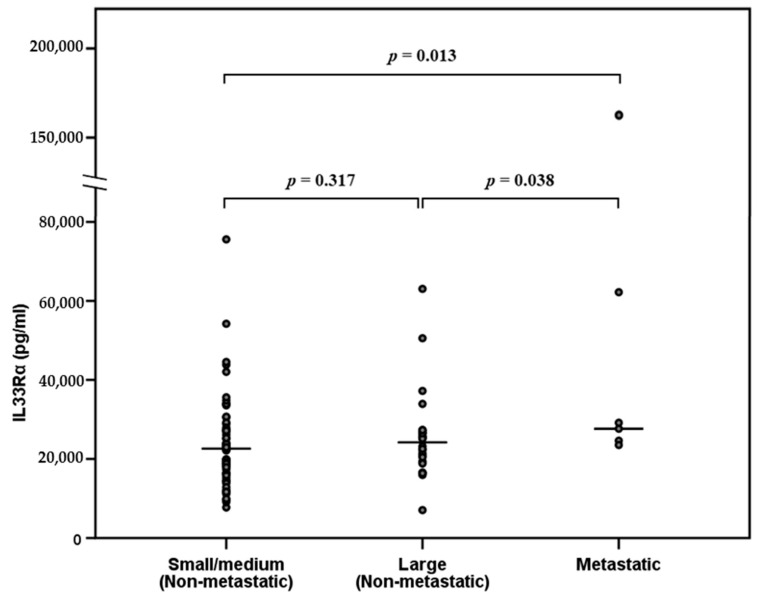
Preoperative IL33Rα serum levels in patients with renal cell carcinoma; a comparison of patients with small tumors (≤7 cm in diameter) with no metastases, with large tumors (>7 cm) and no metastases, and metastatic disease.

**Figure 2 cancers-12-01961-f002:**
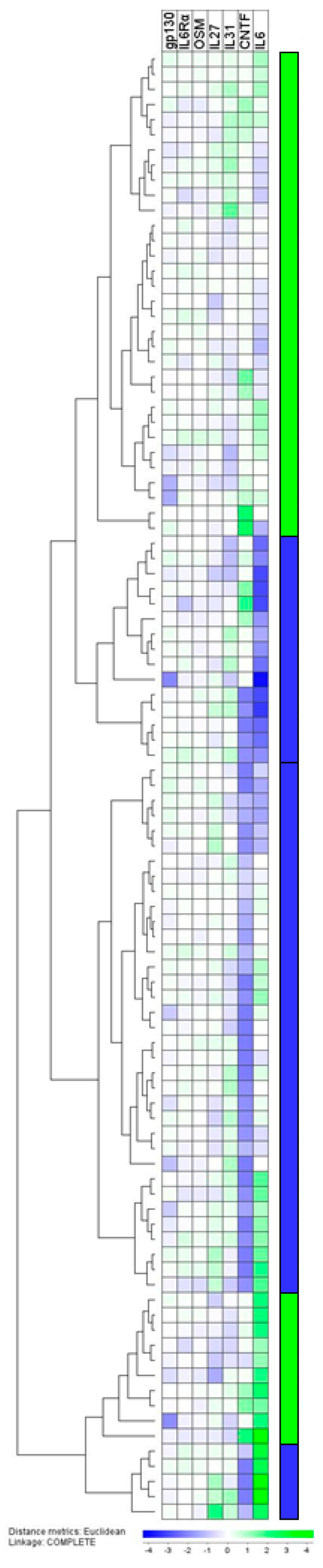
(see page 7). The serum profile of IL6 family cytokines in patients with renal cancer: a hierarchical cluster analysis. This analysis included the seven soluble mediators, IL6, IL6Rα, gp130, IL27, IL31, CNTF, and OSM. Cytokine/receptor is indicated at the top of the figure, and the patient clustering is shown to the left. This analysis created two main clusters (an upper large and a small lower cluster), and each of these two main clusters were further divided into one subset with low IL6/CNTF levels and one with relatively high levels of the two cytokines. Based on these results, we classified the patients into two main subsets referred to as CNTF^high^IL6^high^ (see the right part, indicated by a green color) and CNTF^low^IL6^low^ (right part, blue color).

**Figure 3 cancers-12-01961-f003:**
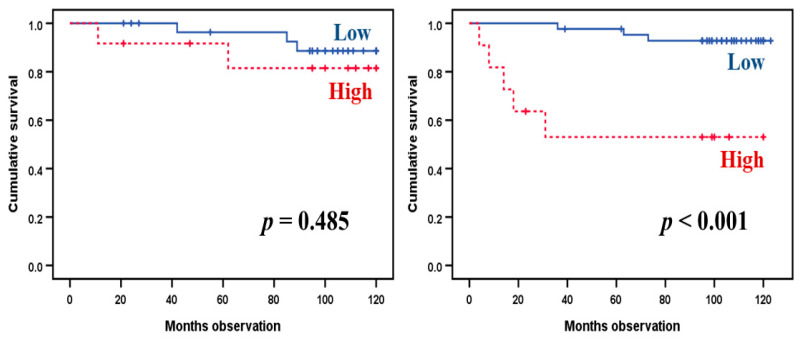
Comparison of kidney cancer-related death for the patient subsets identified in the hierarchical clustering analysis based on IL6 family mediators. As indicated in Figure 2, the 97 patients could be sub-classified into the two main subsets referred to as (left) IL6^high^CNTF^high^ and (right) IL6^low^CNTF^low^ subsets. The patients were classified into quartiles based on their IL33Rα serum levels, and we compared the survival of patients classified in the highest versus the three lowest IL33Rα quartiles. The IL6^high^CNTF^high^ (left) and IL6^low^CNT^low^ patients (right) were analyzed separately. The *p*-values are indicated in the figure images.

**Figure 4 cancers-12-01961-f004:**
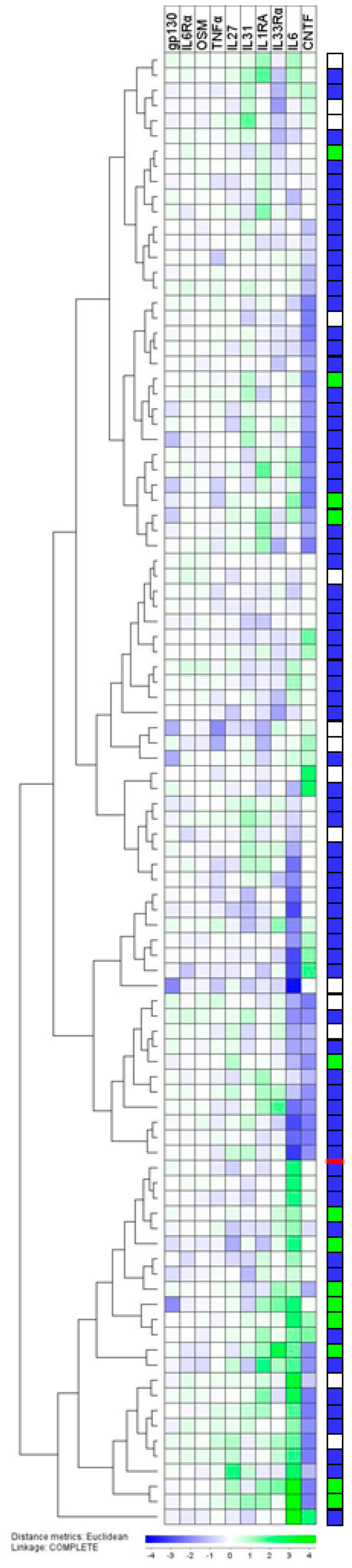
(see page 10). The serum profile of the acute phase cytokines in patients with renal cancer; a hierarchical cluster analysis including IL6 family cytokines (IL6, IL6Rα, gp130, IL27, IL31, CNTF, and OSM), two IL1 cytokine family mediators (IL1RA and IL33Rα), and TNFα. The mediators are indicated at the top of the figure, and the patient clustering is shown to the left. The survival of individual patients is summarized in the right part of the figure and shows patients still alive (blue), dead from renal cancer disease (green), and dead from other causes (white).

**Figure 5 cancers-12-01961-f005:**
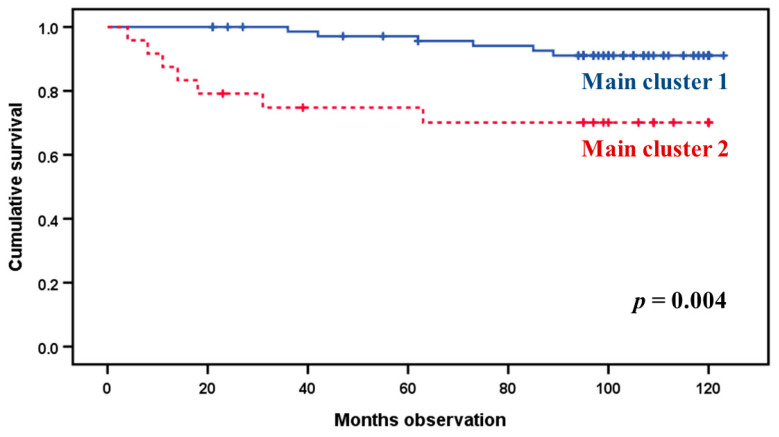
Comparison of cancer-related death for patients in the two main clusters identified in the hierarchical clustering analysis based on the systemic levels of 10 cytokine mediators (see Figure 4; upper main cluster 1, lower main cluster 2). All 97 patients were included in this comparison. A Kaplan–Meier analysis was performed, and the *p*-value for this comparison is indicated in the figure.

**Table 1 cancers-12-01961-t001:** Clinical and tumor characteristics of patients with renal cell carcinoma scheduled for surgery during the period 2007–2010; the table presents the results for all 118 patients for whom a preoperative serum sample was available (unless otherwise specified) and for the patients with local renal cancer disease (i.e., without metastases). The results are presented as the median and interquartile (if specified) range for continuous variables, except for long-term survival. Categorical data are expressed as numbers with a percentage (or in specified cases interquartile range) in parenthesis.

Parameter	All Patients (*n* = 118)	Patients without Metastases (*n* = 109)
Age in years at diagnosis (interquartile range)	63.8 (55.1–2.5)	63.9 (55.4–73.5)
Gender; male/female	88 (74.6)/30 (25.4)	80 (73.4)/29 (26.6)
Charlson Comorbidity Index (interquartile range)	1 (0–2)	1 (0–2)
ASA score (interquartile range)	2 (2–2)	2 (2–2)
Surgical treatment		
Radical nephrectomy	80 (67.8)	71 (65.1)
Partial nephrectomy	38 (32.2)	38 (34.9)
Peripheral blood levels		
B-Hemoglobin (g/dL, *n* = 100/91)	14.2 (8.8–17.3)	14.0 (8.8–17.3)
Erythrocyte sedimentation rate (mm, *n* = 92/84)	13 (2–129)	14 (2–129)
S-creatinine (µM, *n* = 100/91)	76.5 (45–725)	77.0 (45–725)
S-calcium (mM, *n* = 99/90)	2.40 (1.96–3.00)	2.40 (1.96–3.00)
S-alkaline phosphatase (U/L, *n* = 96/87)	81 (45–527)	81 (45–527)
S-CRP (mg/L, *n* = 116/107)	3 (1–220)	3 (1–112)
Tumor size (cm) ^1^	5.3 (1.9–17.5)	4.9 (1.9–16.8)
≤7.0	76 (64.4)	74 (67.9)
>7.0	42 (35.6)	35 (32.1)
Histology		
*Subtype*		
Clear cell	91 (77.1)	83 (76.1)
Papillary	14 (11.9)	14 (12.8)
Chromophobe	6 (5.1)	6 (5.5)
Multilocular cystic	5 (4.2)	5 (4.6)
Others/unclassified	2 (1.7)	1 (0.9)
*Nuclear grade*		
G1-G2	62 (52.5)	62 (56.9)
G3-G4	55 (46.6)	46 (42.2)
Unknown	1 (0.9)	1 (0.8)
Detectable metastases at the time of diagnosis ^2^	9 (7.6)	Not relevant
Observation time (months) ^3^	100 (4–120)	103 (11–120)
Long-term overall survival (mean, standard error) ^4^	96.5 (3.5)	101.7 (3.3)
Long-term recurrence-free survival (mean, standard error) ^4^	106.0 (3.0)	112.3 (2.3)

^1^ Tumor size was measured on CT scans. The complete tumor-node-metastasis (TNM) staging of the patients included in the present cytokine study is given in Appendix A [36,37]. All patients with metastases had tumor diameters > 4 cm. ^2^ Clinical examination together with CT scans of the abdomen and chest were used to classify patients as with or without metastases. ^3^ Patients were observed from the time of diagnosis until death or until November 2018. ^4^ Median survival was not reached.

**Table 2 cancers-12-01961-t002:** Univariate Cox survival predictions in radically treated renal cell carcinoma patients using serum IL33Rα and CRP, as well as the included clinico-histopathological parameters. Values are given as the hazard ratio (95% confidence interval). The whole patient cohort included 109 patients, but the IL33Rα levels were analyzed for 90 randomly selected patients. Other values that differ from *n* = 109 are specified.

Variable	Disease-Specific Survival		Overall Survival	
IL33Rα (ng/mL), *n* = 90	1.05 (1.00–1.09)	*p* = 0.034	1.02 (0.99–1.06)	*p* = 0.178
CRP (mg/L), *n* = 107	1.03 (1.01–1.04)	*p* = 0.011	1.02 (1.01–1.04)	*p* < 0.001
Age	1.05 (0.99–1.11)	*p* = 0.083	1.07 (1.03–1.11)	*p* = 0.001
ASA score	1.43 (0.49–4.19)	*p* = 0.510	1.38 (0.71–2.68)	*p* = 0.342
Tumor size	3.40 (1.58–7.31)	*p* = 0.002	1.66 (1.15–2.39)	*p* = 0.006
Pathological TNM stage	4.53 (2.44–8.44)	*p* < 0.001	2.13 (1.43–3.18)	*p* < 0.001
Fuhrman nuclear grading, *n* = 108	2.51 (1.26–4.98)	*p* = 0.009	1.61 (1.22–2.11)	*p* = 0.001
Leibovich score, *n* = 82 *	4.03 (1.81–8.97)	*p* = 0.001	1.91 (1.19–3.08)	*p* = 0.007

* Patients with clear-cell renal cancer; value missing for one patient.

**Table 3 cancers-12-01961-t003:** The impact of IL33Rα for progression in 67 patients randomly selected out of 83 patients with clear-cell renal cell carcinoma assumed to be radically treated; a multivariate analysis including IL33Rα together with the Leibovich score. The results are presented as the hazard ratio (95% confidence interval) and *p*-values.

Variable	Progression-Free Survival	
IL33Rα (ng/mL)	1.07 (1.01–1.14)	*p* = 0.020
Leibovich, intermediate risk (score 3–5) *	26.9 (2.1–352.0)	*p* = 0.012
Leibovich, high risk (score ≥ 6) *	49.5 (4.3–576.0)	*p* = 0.002
Leibovich, overall	-	*p* = 0.008

* Compared to patients in the low-risk Leibovich group (score ≤ 2), with the maximum score being 11 [38].

**Table 4 cancers-12-01961-t004:** The serum mediator levels in patients with renal cancer; a comparison of the two main patient subsets identified in the unsupervised hierarchical cluster analysis based on the seven IL6 family members (gp130, IL6Rα, IL6, IL27, IL31, OSM, and CNTF), two IL1 subfamily members (IL1RA and IL33Rα), and TNFα. The results are presented as the median level and variation range. The table presents the levels of the IL6 family members included in the clustering analysis together with the levels of IL1RA, IL33Rα, TNFα, and CRP.

Mediator (Concentration)	Upper Main Cluster (*n* = 73)	Lower Main Cluster (*n* = 24)	*p*-Value
gp130 (pg/mL)	92,745 (22,606–121,962)	88,475 (24,351–108,820)	0.332
IL6 Rα (pg/mL)	34,382 (17,789–48,588)	34,057 (22,510–46,610)	0.536
IL6 (pg/mL)	2.9 (0.0–16.3)	↑ 12.1 (0.5–73.2)	<0.001
IL27 (pg/mL)	673 (254–1173)	795 (367–2738)	0.188
IL31 (pg/mL)	196 (87–584)	160 (83–410)	0.058
OSM (pg/mL)	5789 (4500–7911)	5636 (3827–7003)	0.347
CNTF (pg/mL)	454 (98–2555)	274 (98–1961)	0.548
IL33Rα (pg/mL), *n* = 72/24	21,842 (7053–75,572)	↑ 26,652 (15,853–162,569)	0.001
IL1RA (pg/mL)	670 (281–2237)	↑ 876 (488–2711)	0.044
TNFα (pg/mL)	24.5 (6.8–37.2)	↑ 27.8 (18.1–37.9)	0.006
CRP (mg/L), *n* = 71/24	3 (1–19)	↑ 5 (1–220)	0.021

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
