# Peer review of "The Biological Context of C-Reactive Protein as a Prognostic Marker in Renal Cell Carcinoma: Studies on the Acute Phase Cytokine Profile"

_cancers, 2020, doi:10.3390/cancers12071961_

Round 1

Reviewer 1 Report

  1. In this study, the authors proved high serum levels of acute phase protein C-reactive protein is associated with prognosis in renal cancer. All the enrolled patients were received surgical treatment and then assessed the CRP and pro-inflammation cytokines. Although the high correlation between the inflammatory clusters and renal cancer, the pro-inflammatory cytokines are not specific to kidney or cancers. CRP and those cytokines are more significant/apparent in systemic inflammation or other diseases such as in elderly heart failure patients. Patients with renal cancer may also have the other diseases because the average age in patients is elder (63.8-63.9). Were the patients with other diseases excluded? And, how the authors exclude or distinguish the elevated CRP and cytokines being only due to renal cancer but other symptoms?
  2. The English should be polished and concise.

Author Response

  1. Our study is population-based, i.e., we included all patients from a defined geographical area during a defined time period except for a minority of patients where no sample was available due to practical (e.g. not time for blood sampling on the day of surgery) or technical reasons (e.g. sample arriving too late to the laboratory for preparation). Thus, our patient cohort should be regarded as unselected. Even though some patients may have an increase CRP and/or an altered acute phase cytokine profile due to other diseases or inflammaging, the acute phase cytokine profile could still be used for prognostic sub-classification of renal cancer patients. A brief comment is added on page 4. A new chapter is added to the Discussion section.
  2. Spelling and grammar has been carefully controlled.

Reviewer 2 Report

The authors identified among renla cell carcinoma a patient subset characterized by particularly high levels of IL6, IL33Rα and TNFα and an adverse prognosis.

The manuscript is well written and the topic interesting. The authors should report the different stage of renal cancer. 

The authors identified among renal cell carcinoma a patient subset characterized by particularly high levels of IL6, IL33Rα and TNFα and an adverse prognosis. All the enrolled patients were received surgical treatment and then assessed the CRP and pro-inflammation cytokines. The manuscript is well written and the topic is interesting. However, an important aspect correlated with the prognosis and not considered in the paper is the pathological stage and not only the tumor dimension. Therefore, the authors should report the different stage of renal cancers, at least in the table. 

Author Response

We agree that the staging of the cancer disease should be included in the article. This is now presented in Supplementary Table S2. We hope our article based on description of acute cytokine profiles will have a more general interest, and for this reason, we included a description of the various stages in our new Table S2.

The staging data are presented in the new Table S2 as well as in Table 2 regarding its relation to survival, they are referred to in the footnote of Table 1, and commented in a new chapter in the Discussion section.